# HENNOVATION: Learnings from Promoting Practice-Led Multi-Actor Innovation Networks to Address Complex Animal Welfare Challenges within the Laying Hen Industry

**DOI:** 10.3390/ani9010024

**Published:** 2019-01-11

**Authors:** Lisa van Dijk, Henry J. Buller, Harry J. Blokhuis, Thea van Niekerk, Eva Voslarova, Xavier Manteca, Claire A. Weeks, David C. J. Main

**Affiliations:** 1School of Agriculture, Food and Environment, Royal Agricultural University, Cirencester GL7 6JS, UK; David.Main@rau.ac.uk; 2College of Life and Environmental Sciences, University of Exeter, Exeter EX4 4RJ, UK; h.buller@exeter.ac.uk; 3Department of Animal Environment and Health, Swedish University of Agricultural Sciences, 8750 07 Almas Alle Uppsala, Sweden; Harry.Blokhuis@slu.se; 4Animal Health and Welfare, Wageningen Livestock Research, PO Box 338, 6700AH Wageningen, The Netherlands; thea.vanniekerk@wur.nl; 5Department of Animal Protection, Welfare and Behaviour, Faculty of Veterinary Hygiene and Ecology, University of Veterinary and Pharmaceutical Sciences Brno, PalackÉHo TŘ. 1/3, 612 42 Brno, Czech Republic; voslarovae@vfu.cz; 6Department of Animal and Feed Science, School of Veterinary Science, Universitat Autònoma de Barcelona Campus De La Uab Bellaterra, 08193 Cerdanyola Barcelona, Spain; xavier.manteca@uab.cat; 7Bristol Vet School, University of Bristol, Langford, Bristol BS40 5DU, UK; Claire.Weeks@bristol.ac.uk

**Keywords:** practice-led, innovation, networks, laying hen, industry

## Abstract

**Simple Summary:**

HENNOVATION was an EU funded project that aimed to explore the value of networks of laying hen farmers and within the laying hen processing industry, supported by scientists, to improve the health and welfare of laying-hens. During the 32-month project, the project team supported 19 networks in 5 countries and several networks generated new ideas and tested them in their commercial context. The project demonstrated that these networks led by farmers and industry practice can generate practical and effective solutions to animal welfare problems. Greater attention should be given to enhance and support these types of practice-led networks in future strategy and policy initiatives for animal health and welfare improvement.

**Abstract:**

The Hennovation project, an EU H2020 funded thematic network, aimed to explore the potential value of practice-led multi-actor innovation networks within the laying hen industry. The project proposed that husbandry solutions can be practice-led and effectively supported to achieve durable gains in sustainability and animal welfare. It encouraged a move away from the traditional model of science providing solutions for practice, towards a collaborative approach where expertise from science and practice were equally valued. During the 32-month project, the team facilitated 19 multi-actor networks in five countries through six critical steps in the innovation process: problem identification, generation of ideas, planning, small scale trials, implementation and sharing with others. The networks included farmers, processors, veterinarians, technical advisors, market representatives and scientists. The interaction between the farmers and the other network actors, including scientists, was essential for farmer innovation. New relationships emerged between the scientists and farmers, based on experimental learning and the co-production of knowledge for improving laying hen welfare. The project demonstrated that a practice-led approach can be a major stimulus for innovation with several networks generating novel ideas and testing them in their commercial context. The Hennovation innovation networks not only contributed to bridging the science-practice gap by application of existing scientific solutions in practice but more so by jointly finding new solutions. Successful multi-actor, practice-led innovation networks appeared to depend upon the following key factors: active participation from relevant actors, professional facilitation, moderate resource support and access to relevant expertise. Farmers and processors involved in the project were often very enthusiastic about the approach, committing significant time to the network’s activities. It is suggested that the agricultural research community and funding agencies should place greater value on practice-led multi-actor innovation networks alongside technology and advisor focused initiatives to improve animal welfare and embed best practices.

## 1. Introduction

Animal welfare issues such as lameness in cows and feather-pecking in hens are examples of complex, multifactorial challenges that may be better addressed by alternative approaches to the traditional top-down dissemination of knowledge from science to practice. Thus, there is growing policy interest in more ‘bottom-up’, practice-led, collaborative approaches to innovation in Europe [1]. These practice-led approaches respond to the demand for innovation to solve local problems using practical knowledge and creativity at the farm level [2]. Akrich et al. [3] (p. 202), argue ‘the evaluation of the disadvantages and advantages of an innovation is entirely in the hands of the users: it depends on their expectations, their interests, on the problems which they raise’; in short, their practice. Although practical local knowledge is an essential foundation for local innovation, this alone is rarely enough to generate it [4,5,6]. To enable innovation requires creating space for joint learning and knowledge sharing through innovation networks which bring together different actors, with different (forms or sources of) knowledge including science [1,7,8,9]. Klerkx et al. [10] (p. 390) also emphasise the importance of networks for innovation: ‘innovation is considered the result of a process of networking and interactive learning among a heterogeneous set of actors, such as farmers, input industries, processors, traders, researchers, extensionists, government officials and civil society organizations.’

Alternatives to conventional top-down knowledge exchange have emerged. These approaches include bottom-up and joint learning amongst scientists and the farming community [11,12]. These are summarized in Table 1 and described in more detail by Schut et al. [13]. This shift in approach is evident in animal health and welfare initiatives. For example, the top-down advisory approach that was used for a lameness initiative for dairy heifers based on the risk-assessment process commonly used in food processing: hazard analysis and critical control points (HACCP) although providing farm-specific advice to farmers based on the latest scientific research, did not result in much uptake and change in lameness incidence [14]. Thus, a more consultative approach was used in later work to generate improvements in cow lameness [15] and in facilitating improved animal welfare as part of UK assurance schemes [16]. The latter approach also aimed to utilise social marketing principles to maximise the engagement of the farming community. A more participatory bottom-up approach has been promoted within the Stable School methodology [17]. One of the first examples of the use of an Agricultural Innovation System (AIS) approach emerged with the work done by Klerkx et al. [10] on the development of a new laying hen system, the Rondeel system, by a multi-actor innovation network in The Netherlands and the work done using co-innovation platforms for heifer rearing in New Zealand [18]. 

Such practice-led innovation is derived directly from the ‘rooted’ experiences of ‘doing’ their practice, to cope with and adapt to the challenges faced in every-day as well as strategic contexts [19]. Paradoxically, this in-practice and on-farm demand for innovation is rarely seen as a major driving force for applied animal welfare science research. The call for innovation in these practice-led approaches does not emerge from scientific research processes but emerges from the social interactions and the cultural context of individual farmers and their management practices operating within their communities [12,20]. Ultimately, the knowledge of farmers becomes seen more ‘on an equal footing’ with scientific knowledge [21,22]. How to enable these practice-led processes both in impacting upon applied animal welfare science-driven innovation and in delivering practical solutions to improve animal health and welfare on farm by providing relevant/effective science-driven support is a central concern of this paper. 

## 2. The Hennovation Project

Using the egg-laying-hen sector as a case study, the EU H2020 Hennovation thematic network aimed to test mechanisms to enable practice-led innovation through the establishment of networks of farmers and within the laying hen processing industry, supported by scientists. The Hennovation project promoted a multi-directional flow of knowledge, with farmers or industry leading the activity at a local level and scientific researchers and farm advisors supporting the innovation capacity of each network. These networks were established and facilitated to proactively search for, share and use new ideas to improve hen welfare, efficiency and sustainability. A broad definition of innovation was used to include application of both novel and existing ideas or best practices in new circumstances to improve sustainability and/or welfare. 

The policy context for the Hennovation project was the European Innovation Partnership for Agricultural productivity and Sustainability (EIP-AGRI) launched to foster a competitive and sustainable agriculture and forestry that ‘achieves more and better from less.’ A key objective of this strategy was the ‘interactive innovation model’ in which partners with complementary types of knowledge—scientific, practical and other—must join forces in the project activities from beginning to end.’ Hennovation was a thematic network, providing a tool to ‘collect existing scientific knowledge and best practices on the chosen theme and facilitate their use’ and ‘develop easily understandable material for practice, such as information sheets in a common format and audio-visual material’ [23]. The interest in practice-led innovation was driven from the perception of a significant gap between scientific research and the adoption of applied science into farm practice [21]. 

The Hennovation project proposed that solutions to practical problems in the laying hen industry could be practice-led and effectively supported to achieve durable gains in sustainability. The EU laying hen sector was chosen as a case study because of the complexity of the market and legislative contexts of egg production including (1) consumer animal welfare interest in the sector [24], (2) changes in the supply associated with mandatory method of production labelling (Commission Regulation (EC No 589/2008) and (3) legislative changes associated with cage systems (Council Directive 1999/74/EC). In this complex environment, many laying hen producers across the EU were dealing with similar challenges such as the prevalence of health and welfare issues related to injurious pecking [25] and the handling, transport and slaughter of end of lay hens [26,27]. By focusing on practice-led, grass root innovation and its articulation with existing science and market-driven actors, this project aimed to explore the opportunity for such networks to deliver practical solutions within the wider animal production industry. 

## 3. Materials and Methods

The project adapted a participatory action research approach to explore and test mechanisms to facilitate and enhance practice-led innovation through two types of innovation networks in the laying hen sector; local on-farm networks led by groups of producers and national and (inter)national off-farm networks led by hen processors and industry. These two ‘models’ of innovation networks (on-farm and off-farm) were selected to explore the roles of different actors (farmers, veterinary surgeons, farm advisors, scientists, egg buyers, certification schemes), different scales of networks (small farm group versus larger industry network) and different external drivers (legislation, consumer pressure, market forces, productivity and profitability). Prospective network members in the five participating EU Member States (UK, Sweden, The Netherlands, Spain & Czech Republic) were recruited by direct contact or via the existing collaborative networks of the project partners. 

Stimulating a dynamic practice-led innovation process was expected to: (1) identify innovation needs and potential solutions (2) exchange knowledge and seek consensus on needs and solutions (3) test technical and economic viability of solutions and (4) lead to wider application and sharing of solutions [28]. The research methodology, based on Moschitz et al. [8], included a reflection and action process at the facilitators’ level and a co-learning process at the network level. A facilitators’ coordinator supported the facilitators and acted as ‘reflexive monitor,’ probing the way the facilitators worked and their underlying assumptions through reflection workshops [18,29].

In total 11 facilitators from the five participating countries were recruited to support the innovation networks. All had university degrees related to animal health and welfare or veterinary sciences with experience of working in the livestock sector; some had experience of the laying hen sector. The facilitators had a varying degree of experience in collaborative research projects, some had little or no previous experience whilst others had facilitated more collaborative research processes, though not necessarily focusing on innovation by farmers. The facilitators developed the skills and techniques of facilitation by participating in the exercises during four facilitator-focused workshops arranged at critical stages of the project. Tools such as network mapping, Venn diagram for stakeholder analysis and the learning history [30,31], were used to monitor network performance and self-reflection by facilitators, with the idea that the facilitators could also use these tools for self-reflection with their networks.

A framework for the adaptive management and facilitation of practice-driven innovation was developed by and with facilitators during the first facilitators’ workshop. Through a series of workshop exercises, the facilitators charted the distinct stages, or ‘process steps’ towards innovation. This on-going process enabled facilitators to develop a shared understanding of an innovation network’s support needs and effectiveness, that is, ‘network health’ [32]. Initially the facilitators identified six key steps in the innovation process:
The identification of the need for innovation; shared problem/opportunity,The generation (and assessment) of innovative ideas which could provide potential solutions,The selection of an innovative idea and plan of action to ‘test’ the idea including resources required in terms of time, external expert support and money,The practical ‘testing’/development of the idea on-farm, during transport or at the slaughter house,The implementation and upscaling of the innovation in practice,Finally, the wider dissemination of the innovation amongst the sector.

Although the framework is presented stepwise, the innovation process is rarely linear and the time allocated for each step cannot be predicted [33]. Moreover, it depended, amongst other things, on a variety of other factors such as network capacity and the nature of the intended innovation itself. The challenge in the development of the framework was that on the one hand it needed to provide enough structure to be useful for the facilitator whilst on the other hand it needed to be generic and flexible enough to accommodate the diversity and unpredictability of the process [33,34]. For example, the practical trialling and development description (step 4) could have included small scale experimental trials or a more informal feasibility trial. Further discussion of each step, within the facilitation workshops, led to the development of more detailed guiding questions/criteria (Table 2). These were developed as a tool to guide the facilitation of the innovation process, record level of engagement (Figure 1 and Figure 2) and to stimulate facilitator learning in managing the process in the field. In addition to individual reflection by each network facilitator, participants were also asked to compare the progress and functioning of each network and identify why similarities or differences in performance emerged.

An essential role of the network facilitator, apart from guiding the network through the innovation process and promoting learning, was to link with different support actors including scientists. The involvement of science-driven support actors such as applied animal welfare scientists, veterinarians and technical advisors was an essential part of the innovation process combining different types of knowing and creating a diversity of knowledge [20]. It was envisaged that scientific knowledge was brought to the network based on the network’s demand for this knowledge at any particular stage in the innovation process. 

## 4. Results

### 4.1. Recruitment and Description

The project recruited a total of 19 multi-actor networks across the Czech-Republic, The Netherlands, Spain, Sweden and the United Kingdom. Fifteen networks focused on finding practical solutions to problems related to feather (or injurious) pecking on-farm. Four further networks focused on practice-led innovation on the handling and use of end-of-lay hens (off farm). Depending upon the time taken to recruit groups the networks were active for up to 18 months during the project. Most networks held face to face meetings although some networks occasionally used conference calls to overcome the logistical challenges of a geographically dispersed network. The project team avoided pre-defining the term ‘network’ to allow for various routes to network formation [34]. Networks were based upon pre-existing farmers group (*n* = 2), or pre-existing larger groups connected to egg packer or veterinary practices (*n* = 8) or were newly established as part of the project (*n* = 9). The network size of the on-farm networks varied from three to 25 members (Table 3). There were 124 active members participating at some stage in the 19 networks although for some networks the numbers that attended each meeting varied. Six networks included at least some organic farmers, 3 networks included some producers with furnished cages and the majority (12) included farmers with free-range or barn (aviary) systems. In several on-farm networks, apart from farmers, other network actors, such as veterinarians and field support staff of the egg packers, were part of the actual network. Whilst in other on-farm networks these actors were not part of the actual network but provided specific support relevant for the topic addressed by the network. The four end-of-lay networks included actors (such as production, catcher and abattoir managers, hen processors, handling equipment manufacturers, industry organisation representatives) actively involved in the process of depopulating, transporting and use of hens at the end of their laying period.

### 4.2. Network Activity

The Hennovation networks tackled a range of technical challenges by developing different types of innovations. A few ideas were radical, yet the many which were incremental served to increase motivation and build the capacity of a network to innovate. Alongside technical ‘hard’ or product innovations (e.g., new type of litter material to reduce stress and encourage natural behaviour or the use of alpacas in organic systems to reduce predation), a variety of often less expected and sometimes unintended ‘soft’ innovations (i.e., processes and protocols) also emerged through these networks (e.g., a new way of marketing low value hen meat and new relationships between production chain actors such as pullet rearers). The process led to innovation on both individual and collective network levels. Some ideas developed and tested were innovative in a specific farm context (for example the use of different range covers, sheds, cover crop and trees to encourage birds out onto the range) though not necessarily innovative for the laying-hen sector. Others had a potential to have a great impact on the sector (for example the use of trolleys when catching hens and immediately placing them into the drawers in which they are transported to the processing plant). 

Facilitators used the framework to reflect on the functioning of their networks as they moved through the innovation process, which they presented during the reflection workshops in project month 17 (May 2016), month 23 (November 2016) and month 29 (May 2017) (Table 2 and the final scoring reported during the facilitator’s workshop in month 29 is presented in Appendix A). 

The facilitators were asked to reflect on whether intervention was required, on what level and what kind of intervention was necessary to move the innovation process of the network forward. Examples of interventions required included the involvement of a scientist with specific knowledge on a problem identified by a network to jointly find solutions to in step 1 or the involvement of farm business advisor to assess the feasibility of an idea developed in step 3. Although the scoring (High, Medium and Low) between the facilitators was not standardised, this basic summary of the results across the networks provides some insight on the perceived functioning of networks by the facilitator (Figure 1 and Appendix A). The facilitators perceived that both the level of enthusiasm and energy of the network members and the trust and knowledge sharing between network members were generally high (14/19 networks). Considering the high level of participant engagement, it is not surprising that the level of facilitator intervention required was relatively low, with only four of the 19 networks requiring high levels of facilitator intervention. 

In the last reflection workshop in project month 29, the network facilitators and social scientists reflected upon the factors that were promoting effective practice-led innovation. The network facilitators discussed and listed 13 enabling factors: shared opportunity, motivation, knowledge, trust, collective purpose, contacts within the poultry sector, capacity within the production system, market and legislative ability [27].

Analysis of the progress of each network on the six steps shows that in project month 29 (May 2017), all networks were active, with over 90% (*n* = 19) completing the action planning (step 3, see Figure 2). For those networks that had completed or were working at a particular step, there was generally high engagement. There was some indication that the financial feasibility of the innovation was not always valued with only five networks assessed as using the cost-benefit calculator provided by the project (step 2.2 in Table 2). Ten networks were scored as having at least some satisfaction with the “relevance and affordability of solutions developed” and “pride in what they achieved,” although only five networks were deemed to be applying the innovation as “common practice” across the members of the network. Despite the relatively short time scale of the project (32 months), seven networks actively sought to disseminate innovation beyond network members (step 6.1 in Table 2).

As expected for a project with a limited time scale the extent of the trials varied between networks. For example, five networks were able to investigate the feasibility of an innovation with a further two networks also able to design a detailed trial to examine an innovation. Nine networks undertook pilot studies generating initial results, with a further three networks producing results that were expected to be publishable in a peer-reviewed journal. In terms of the agreed next steps within each network, six networks believed there was sufficient evidence to influence future management decisions. Nine networks wished to continue further trials with a further two networks keen to seek further financial support for trial work beyond the end date of Hennovation. 

The practice-led innovation approach aimed to reduce the gap between science and practice thus scientific knowledge and information was introduced to the innovation process in several unconventional ways. Some facilitators supported the network in doing a background scientific literature review such as in the case the UK Network 3 which required information on hens perceive light; some facilitators shared scientific journal articles with the network members; and other facilitators summarised an area of science into short, practical summaries for their network. In many networks, the facilitator brokered expertise through inviting a scientist or technical advisor to discuss a specific topic of interest such as in the case of the NL Network 13 and the ES Network 8. The different strategies for integration of science had varied impact. Some facilitators indicated it was quite challenging to make the scientific information relevant for the network. Some networks indicated they valued gaining the tailored scientific information which could be applied on farm whilst others did not find this as useful. Where trial work was undertaken, relevant outcome measures were used to assess the impact of innovations. These included productivity parameters (e.g., egg production and mortality), welfare measures (e.g., feather scores and Welfare Quality^®^ protocol(s) parameters) and environment indicators (e.g., ammonia and presence of red mite). One or more individuals in each group tended to take the lead in providing scientific support. For at least nine networks the facilitators provided this scientific support. For seven networks another researcher with specialist expertise provided this input. For other networks a range of individuals provided scientific support including veterinarians, consultants, advisors and suppliers. Interestingly, the interaction of the networks with these support actors, specifically the scientists, created a mutually beneficial relationship which in some countries has led to new working practices between scientists and farmers based on experimental learning and co-production of knowledge. In the Czech Republic, the university involved in the project has decided to continue to support the farmer-led trials and in Spain they decided to continue using the practice-led approach to promote innovation to improve welfare of pigs.

### 4.3. Innovation Support Actors

Most networks involved support actors at some stage during the project (Table 2). In all, 66 support actors were involved across the 19 networks. These included market actors (egg packer, certification inspectors or advisors), supply chain actors (nutritionists, pullet rearer, hatcheries), technical experts (lighting technology, parasites, hen behaviour), poultry advisors, veterinarians, government representatives, industry representatives and university researchers. The support actors involved in the networks undertook the following broad roles:
(a)Industry support actors brought legitimacy to the practice led innovation and connectivity within the industry to enable the upscaling of innovation and relevance beyond its original development;(b)Scientific support actors brought expertise, conceptual understanding and normative knowledge to the networks to facilitate the development of practice-led innovation and (c)Technical support actors brought technical material and practical resources to the network to enable ideas to be translated to workable solutions.

External support actors varied in the level of their engagement with the networks. Some worked predominantly through the facilitator without attending network meetings (for example, a Dutch scientist provided a webinar on lightening for UK Network 3). Others were more closely involved, attending meetings and offering network farmers advice and assistance directly. Some facilitators (for example, the facilitator working in the Czech-Republic) considered their own home institutions (in most cases a University) as a support actor rather than a network member. Off-farm end of lay networks differed from the on-farm networks in the structure of their support. For these, the networks themselves comprised a variety of industry actors. The NL Network 17, for example, comprised an ‘inner circle’ of industry actors who met regularly and an ‘outer circle’ of further industry actors who were involved less directly in the network. Support actors were invited by network members directly or invited via the network facilitator and these actors generally supported the networks on a voluntary basis. Some came in once to provide specific knowledge, others worked with the network for longer; for example, a veterinarian supported a network during data collection in The Netherlands.

## 5. Discussion

Promoting innovation in agriculture is and will remain a major policy priority considering the emerging challenges of climate change, anti-microbial resistance and changing consumer demand in a growing population. The multi-actor practice-led innovation networks that have been supported in the Hennovation project have demonstrated that a practice-led approach involving market and science-based actors can be a significant stimulus for innovation and make a meaningful contribution to these challenges. Several networks generated novel ideas and tested them in their commercial context. Alongside material or technical innovations (such as a new trolley design for depopulation of hen or the use of alpacas in organic systems to defend hens against predators), a variety of often less expected and sometimes unintended ‘soft’ innovations also emerged through these networks. These were related to protocol or process (e.g., a new way of monitoring Poultry Red Mite infestation and new relationships between value-chain actors, for example between pullet rearers and egg producers). The complexity and the novelty of some of the innovations was significant. The farmers and processors involved in the project were often very enthusiastic, committing significant time and in some cases financial investment to the group’s activities. This is well summarised by the Hennovation Project Advisory Board’s comment that ‘The focus on producer-led innovation is what makes the project unique and important, bridging the gap between policy and science and producer and industry needs and realities.’ The board suggested that ‘The Hennovation project shows there is interest for producer-led innovation and that the industry is willing to contribute their time and effort to engage in both the facilitation process and the development and implementation of innovative practice.’ 

The practice-led approach used here is not a new concept. A ‘farmer first’ approach was advocated for those supporting resource-poor agriculture in developing countries [35]. The ‘farmer first’ approach was a participatory methodology where farmers were supported to (a) analyse (b) consider choices and (c) to experiment on their own situation. Outsiders (or external actors as described in this paper) were encouraged to support these three stages by acting as (a) ‘convenor, catalyst, advisor’ (b) ‘searcher, supplier, travel agent’ (c) ‘supporter and consultant.’ These descriptions of the farmer and outsider roles could well have been used to summarise the activities of the network members, facilitators and support actors in the Hennovation project. The proponents of the farmer first approach suggested that a participatory approach was necessary in the resource-poor production systems where ‘simple and high input packages do not fit well with the small scale, complexity and diversity of their farming systems.’ Our suggestion is that current livestock production in the EU also needs more than simple solutions. The complexity of each farm and their production system, producing products for very different markets means that advisory services must adopt a more flexible approach.

Practice-led innovation processes were network specific and evolved as the actors within the network came together to share common problems, experiment with possible solutions and learn. Van Dijk et al. [36] p. 4 describes the factors enabling practice-driven innovation in the laying hen sector identified by the network facilitators; conditions for innovation to happen (e.g., shared opportunity, motivation and knowledge), conditions to work effectively as a network (e.g., trust, collective purpose and contacts) and conditions for successful application in practice (e.g., capacity within the production system and market and legislative ability). In broad terms, however, it was the engagement, enthusiasm and expertise of the network members that is the fundamental resource for agricultural innovation. Realisation of this potential requires active participation from relevant actors, professional facilitation, moderate resource support and access to relevant expertise.

The innovation networks received science-driven support and many networks valued the contribution of tailored scientific and technical knowledge to help address their specific husbandry challenges. The interaction between the farmers and scientists (along with other actors such as veterinarians and technical advisors) was seen as important for farmer-led innovation to occur. New relationships emerged between the scientists (science) and farmers (practice) based on experimental learning and the co-production of knowledge for improving animal health, welfare, productivity and sustainability. In this way, the relationship between science and practice moved away from the more instrumental researcher-farmer relationship to a more collaborative one working in co-innovative partnerships to jointly develop local integrated innovations for complex problems. The role of the scientist changed to supporting the innovation process, often with an enabling or facilitating role to providing scientific knowledge only as required by the needs of the network. Innovation facilitation required different skills including curiosity in understanding the complexity of husbandry challenges and market requirements, confidence in their ability to support an evidence-based process and sufficient humility to recognise the value of practical experience. This facilitating role was at times quite challenging for the scientists as it requires a shift in attitude and new experiences on the part of the scientists, not necessarily those gained through their academic career. 

The results show that promoting more participatory approaches to farmer innovation leads to a more complex interaction between science and practice. In the Hennovation project the practice-led innovation approach did not prioritise either knowledge of farm practice or science. Rather, it acknowledged the joint contextual knowledge developed through the innovation process, which emerged as a collective rather than individual property. The practice-led approach was based on problem-solving and was responsive and generative, rather than reductive. This was substantially different from the traditional development process of agricultural innovation, which is often reductive and largely generated at a distance. In these alternative approaches, scientists are directly involved and become part of the innovation networks, working together with the farmers to address their needs while ensuring the research is relevant to the farmers’ practices. One might hope that such experiences will allow researchers to better link their own future research projects to the real needs of farmers. Although informed by science, practice-led innovation was not wholly science-driven but was empowered by the need of multiple-actors to find innovative solutions in and through practice, as can be seen from for example of the market-led work done by SE network 17 on promoting the use (consumption) of End-of-Lay hen meat to increase its value and public demand. Hence the Hennovation innovation networks contributed to bridging the science-practice gap by jointly finding new solutions using science to support rather than dictate the needs of practice. 

The scientists involved in the Hennovation project welcomed the change in their role to a less formal, more practically relevant and closer working relationship with practitioners. Their involvement, however, was sometimes seen as challenging as within their research institutes they are increasingly required to demonstrate tangible and recordable outputs and clear unequivocal impact. The innovation process was no longer solely under their ‘control’ leading to challenges to the procedural rigor required in conventional science. Who ‘owns’ the knowledge was also far less straightforward. The innovative ideas identified by farmers (for example on the use of lighting and ozone in hen houses) revealed considerable opportunities for practice-led innovation processes to create valuable outcomes credible to science. Yet, this type of innovation process is not always accorded the scientific legitimacy and value it deserves given the potential it offers for high(er) research impact pathways and for valuable interchange through the active engagement of researchers in co-innovation. Within the scientific community, new ways to value this form of research and engagement, we argue, would be required to realise the potential for this approach.

## 6. Conclusions

Practice-led approaches, as developed and examined in the Hennovation project, are not a universal panacea to achieving innovation in farm animal welfare. The results of the Hennovation project demonstrate that such approaches have their place, yet they are arguably less effective where issues are more straightforward and regulatory solutions are appropriate and they will not replace the need for traditional science-led initiatives. What we can argue however, is that a greater value should be placed on and greater attention given to, these participatory approaches to practice-led innovation alongside more conventional innovation pathways and other welfare improvement strategies particularly in addressing complex, multi-factorial issues. More opportunities are needed to enhance the integration of such participatory approaches to practice-led innovation in future strategy and policy initiatives for animal health and welfare improvement. In particular policies should be developed that (1) promote the role of the facilitator such as specific innovation facilitator training and support within relevant agricultural and scientific institutions; (2) highlight the value of social capital within local farming communities to support and encourage innovative solutions to real-world practical issues; (3) enable access to relatively small amounts of funding for farmer networks seeking to trial innovative activities or procedures; and (4) encourage the establishment of partnerships between industry, science and technical actors to help generate cooperative and co-innovative partnerships with farmers.

## Figures and Tables

**Figure 1 animals-09-00024-f001:**
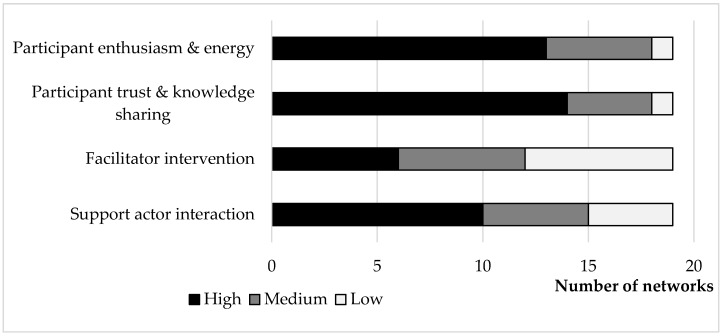
Overview of the scoring by the network facilitators of the general indicators of the functioning of the Hennovation innovation networks in project month 29 (May 2017).

**Figure 2 animals-09-00024-f002:**
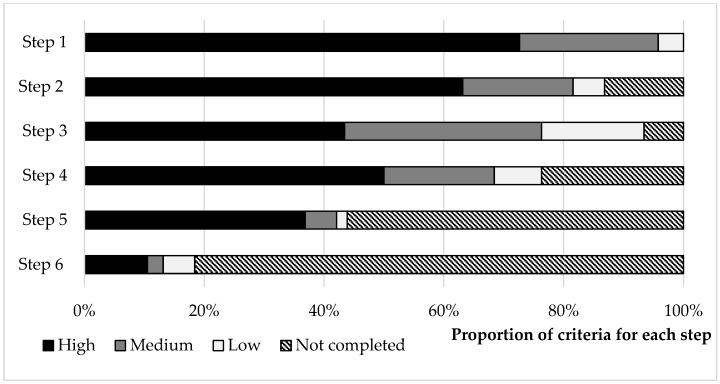
Proportion of innovation networks (*n* = 19) scored as achieving high, middle and low engagement for the criteria associated with each of the six facilitation process steps at the completion of the project. The number of criteria ranged from 2 to 4 for each step (see Table 2 and Appendix A).

**Table 1 animals-09-00024-t001:** Overview of the four approaches to agricultural innovation, (based on [13]).

Approach	Transfer of Technology (TT)	Farming Systems (FS)	Agricultural Knowledge and Information Systems (AKIS)	Agricultural Innovation Systems (AIS)
Originally proposed	1950s–1980s	1980s–1990s	1990s–2000s	2000s–onwards
Key-objectives	Transfer, diffusion and adoption of technology	Contextualise agricultural research and technology	Build local capacities, empower farmers	Enhance systems capacity to generate and respond to change
Flow of innovation	Top-down	Top-down	Bottom-up	Multi-directional
Key intervention approach	Technology dissemination through extension Use mass media to facilitate adoption	Farmer consultation to inform research Surveys to develop farm typologies, modelling of innovation impact	Conduct participatory research Implement joint learning activities	Establish and implement multi-actor innovation platforms
Role of farmers	Adopters of technologies	Adopters of knowledge and technologies Source of information	Experimenters Experts	Partners Entrepreneurs Part of innovation network
Role of research and researchers	Developers of knowledge and technologies	Experts	Capacity builders Facilitators of learning	Actors to enhance innovation capacity in the system Members innovation network
Example of animal health & welfare initiatives	Heifer Lameness Control Plans [14]	AssureWel, [15]. Healthy Feet Project [16]	Stable Schools [17]	Hennovation Thematic network (reported here)

**Table 2 animals-09-00024-t002:** Framework to support facilitation of practice-led innovation processes (source [28]).

Overall General Engagement	
**Step 1 Problem identification**	
1.1	Level of clarity of purpose and shared objective as a network
1.2	Level of agreement on network function (e.g., decision making, common rules, reaching consensus etc.)
1.2	Degree to which the problem identified was based on shared need (common problem)
1.3	Degree to which market or other actors value the problem (relevance)
1.4	Capacity of network to find practical solutions to the problem identified (perceived capacity of the network by the facilitator)
**Step 2 Generation of ideas**	
2.1	How strongly the idea/solution is shared by members of the network
2.2	Feasibility of the idea (includes financial viability, based on ADAS tool)
2.3	Level of diversity of knowledge (resources) used: science, advisor’s and other actors’ input, practical experience etc.
2.4	Capacity of network to trial the practical solutions selected (perceived capacity of the network by the facilitator)
**Step 3 Action planning & resource mobilization**	
3.1	Robustness of innovation action planning including time-frame and task division (everyone knows what is happening, when and by whom)
3.2	Level of clarity on anticipated result (research question) and system or criteria in place to measure and monitor the results (i.e., viability)
3.3	Level of resources the network members commit towards trialling.
3.4	Level of external support (whether scientific, from industry or technical)
**Step 4 Practical trialling and development**	
4.1	Level and rate of innovation—action plan leads to action.
4.2	Willingness to discuss and share within the network successes and failures (to learn from failures)
**Step 5 Implementation and up-scaling**	
5.1	Level of satisfaction of members with regard to relevance and affordability of solutions developed.
5.2	Number of network members applying the innovation as common practice in their business
5.3	Network members’ pride of what they achieved (wanting to share and scale -up the innovative idea).
**Step 6 Dissemination**	
6.1	Network has actively sought to disseminate innovation beyond network members
6.2	Innovation has been subsequently adopted by other actors and bodies

**Table 3 animals-09-00024-t003:** Overview of innovative ideas tested by the on-farm & off-farm innovation networks.

#	Network Location *	On-Farm/Off Farm Solution Tested	Active Members	Support Actors
1	UK	Different range covers (shelters, cover crop and trees) to encourage birds out onto the range.	5	3
2	UK	Sand as an alternative litter substrate to reduce stress and increase natural behaviour and consequently reduce of injurious pecking.	7	4
3	UK	Different colours LED lighting in different areas of the shed to promote natural behaviour and reduce feather pecking.	5	4
4	UK	Ozone treatments of the air in the sheds to reduce poultry red mite infestation.	7	4
5	UK	The use of a probiotic to improve the gut-health of laying hens.	6	4
6	ES	The variation in amino acid levels in different food batches and how this affect laying hens.	7	1
7	ES	Compare different ways to measure ammonia in stable and how the climate and management routines affect the air quality.	4	3
8	ES	Simple low costs traps to monitor the development of Poultry Red Mites in cage systems and develop a monitoring protocol.	6	5
9	ES	Alpacas on farm within an outdoor range to reduce the number of attacks on hens from predators.	5	5
10	CZ	Various discussions on management factors including nutrition, flock density, red mites and an innovative substrate to reduce wet litter.	3	2
11	CZ	The use of new biocide spray against red mites, which has been introduced to Czech market.	4	5
12	NL	The use of different litter type, cut rapeseed straw and cut fibre hemp and its effect on hen behaviour, feather pecking, animal health and red mites.	6	4
13	NL	The most favourite pecking block for laying hens with the highest longevity and efficacy	11	1
14	NL	Lighting characteristics on farms with different lighting systems, including light spectrum and the effect on egg shell quality in laying hens with intact beaks early in the laying period.	25	13
15	NL	Practical protocol for Integrated Pest Management (IPM) of poultry red mites (PRM)	3	1
16	UK-EoL	The use of trolleys to load birds at their ‘home’ cage in the house and wheel this outside to transfer the drawers full of birds into the transport module in order to avoid carrying birds upside down.	6	3
17	NL-EoL	Supplementing drinking water with a heat-stress reducing product a day before end of lay.	7	1
18	SE-EoL	Promoting the use (consumption) of End-of-Lay hen meat to increase its value and public demand.	5	1
19	CZ-EoL	More frequent replacement of lids on transport crates to reduce injury rates (broken bones and bruising) in transported hens.	2	2

* Project country abbreviations, no 16 to 19 are End-of-Lay (EoL) off-farm networks.

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
