# Peer review of "HENNOVATION: Learnings from Promoting Practice-Led Multi-Actor Innovation Networks to Address Complex Animal Welfare Challenges within the Laying Hen Industry"

_animals, 2019, doi:10.3390/ani9010024_

Round 1

Reviewer 1 Report

Title: HENNOVATION: learnings from promoting practice-led multi-actor innovation networks to address complex animal welfare challenges within the laying hen industry

General comments:

 This sounds like a very worthwhile project, and initiatives such as this are certainly important. Well done for carrying out what sounds like a quite challenging undertaking. Unfortunately, at this point the paper doesn’t really do the project justice. The writing is long and complicated. Sometimes it is very hard to follow with many sentences having to be read several times. As a priority, I would recommend carefully going through the paper to make the writing much more clear and concise. Each sentence in this paper could probably be reduced by a third of the words and it would both improve the clarity of your meaning and the ease of reading it. I think this is particularly important for a paper that is discussing engagement of different groups into animal welfare initiatives, and if published in Animals it will be open access so it will not only be read by scientists. My point is that the language needs to be made much accessible to readers who can make use of your work.

Furthermore, the structure of the paper itself needs to be tackled in a more methodical way. Ensure that the content is in the correct sections (e.g. methodological detail was in the results) and that right from the beginning of the introduction the reader understands what issue you are trying to address.

Specific comments (non-exhaustive):

·      L53. There needs to be a strong opening statement I believe on the problem you are trying to address (which I read as the gap between science and practice), as the reader needs to get through a large amount of information before they get to anything on animals and I suspect you will lose a lot of people along the way.

·      L62. “390” should be “p390”

·      L67. Reference needed after “experimental research design”

·      L74: The sentence starting at this line is hard to follow

·      L75. Comma after “1960s”

·      L79. Full stop needed after [15]

·      Be careful about ensuring the concepts are adequately explained. For example, at L84 AssureWel and Healthy Feet Projects are mentioned, but are not explained. There are probably well known to some readers but definitely not all readers.

·      The word “more” is favoured throughout and becomes quite distracting, especially when it is written three times in a single sentence.

·      L81. Remove “previous” as it isn’t yet evident that you are discussing animal health and welfare initiatives

·      L82: What are HACCP principles? This hasn’t been introduced previously and is not a self-explanatory term.

·      L97: Reference needed after “communities”

·      L102: closing bracket needed after [15]

·      Table 1: Might be useful to have a row that is for relevant references for each column

·      L114: Perhaps insert “industries” after “forestry”?

·      L117: This direct quote in ‘ ‘ would be better paraphrased. Similarly for the sections that follow it.

·      L134: Reference needed after “end of lay hens”

·      L141: This paragraph should probably be combined in the methods under a ‘Participant’ section

·      At the end of the introduction it would be good to make the research aims/goals/questions very clear

·      L163. Facilitator training is an important part of this work and so needs more detail on what was delivered, when and how

·      More detail is needed on how the facilitators actually facilitated the two networks. How were they assigned? What was measured when they interacted. Etc etc.

·      L201: The Mathe sentence seems an abrupt way to end this section

·      L206: These networks were of which type as described at line 141?

·      L206: A lot of this should be in the methods section

·      L216: table needs a capital Table

·      L229: from “equally valued…” this needs some evidence. How was this measured within networks?

·      The results section needs to be better detailed in a more methodical way, preferably using some figures to detail the responses by networks on the scoring. This will also make it easier to determine whether the discussion draws appropriate conclusions from the results.

Author Response

The authors appreciate the response from the reviewer, who clearly sees the manuscript as worthwhile and important. We have taken on board the reviewer’s comments and have changed the manuscript substantially in order to address these.

Point 1: An extensive edit of the English language and writing style has been carried out to make the writing clearer and more concise.

Point 2: The structure of the paper has been revised; content moved to the correct section and detail added in the results section by revising the table and by adding two figures presenting the results.

Point 3: Specific comments made by the reviewer related to references, need for closing brackets and punctuation etc (L62, L67, L74, L75, L79, L81, L97, L102, L117, L134 & L216) all have been corrected.

Further responses to specific comments:

Point 4, Reviewer’s Comment L53: There needs to be a strong opening statement I believe on the problem you are trying to address (which I read as the gap between science and practice), as the reader needs to get through a large amount of information before they get to anything on animals and I suspect you will lose a lot of people along the way. Authors’ response:  Many thanks to the reviewer for pointing this out, a very valid point. We have added a clear opening statement.

Point 5, Reviewer’s Comment: Be careful about ensuring the concepts are adequately explained. For example, at L84 AssureWel and Healthy Feet Projects are mentioned, but are not explained. There are probably well known to some readers but definitely not all readers. Authors’ response: We have deleted the direct reference to these projects and included ‘Thus a more consultative approach was used in later work to generate improvements in cow lameness [15] and in facilitating improved animal welfare as part of UK assurance schemes [16].

Point 6, Reviewer’s Comment: L82: What are HACCP principles? This hasn’t been introduced previously and is not a self-explanatory term. Authors’ response: We have revised this sentence to explain these and now reads as follows:

‘For example, the top-down advisory approach that was used for a lameness initiative for dairy heifers based on the risk-assessment process commonly used in food processing: hazard analysis and critical control points (HACCP) although providing farm-specific advice to farmers based on the latest scientific research, did not result in much uptake and change in lameness incidence [14].’

Point 7, Reviewer’s Comment L141: This paragraph should probably be combined in the methods under a ‘Participant’ section. Authors’ response: This paragraph has been moved to the Methodology section

Point 8, Reviewer’s Comment: At the end of the introduction it would be good to make the research aims/goals/questions very clear. Authors’ response: There is a clear statement at the end of the introduction about the research aims and this is further elaborated on in the description of the research project, section 2

Point 9, Reviewer’s Comment L163: Facilitator training is an important part of this work and so needs more detail on what was delivered, when and how? Authors’ response: The authors completely agree that the role of the facilitator is crucial in this type of innovation processes and facilitation training is important. However, this was not the subject of this manuscript. Details about facilitating this process has been published in van Dijk, L.; Buller, H.; MacAllister, L.; Main, D. Facilitating practice-led co-innovation for the improvement in animal welfare. Outlook on Agriculture 2017, 46, 131-137. https://doi.org/10.1177/0030727017707408

Point 10, Reviewer’s Comment L201: The Mathe sentence seems an abrupt way to end this section. Authors’ response: The authors reviewed this comment and decided to remove this sentence.

Point 11, Reviewer’s Comment L206: These networks were of which type as described at line 141?: Authors’ response: As described in the text following L206: ‘Fifteen networks focused on finding practical solutions to problems related to feather (or injurious) pecking on-farm. Four further networks focused on practice-led innovation on the handling and use of end-of-lay hens (off farm).’

Point 12, Reviewer’s Comment L206: A lot of this should be in the methods section.  Authors’ response: We considered this suggestion; however, this paragraph presents the results of the activities generated by the innovation support process described in the methods. Hence, we perceive this is not part of the methodology but part of the results.

Point 13, Reviewer’s Comment L229: from “equally valued…” this needs some evidence. How was this measured within networks? Authors’ response: We thank the reviewer for pointing out that this statement was not sufficiently evidenced, and we changed the sentence to: ‘A few ideas were radical, yet the many which were incremental served to increase motivation and build the capacity of a network to innovate.’

Reviewer 2 Report

In their paper the authors present different strategies of applied agricultural research. They discuss one novel and promising approach, i.e. Agriculture Innovation Systems, on the basis of the Hennovation project in detail. To effectively improve the life of farmed animals in terms of health and welfare, such new concepts exceeding the constraints of “classical science” are absolutely necessary. The present approach is a valuable example on how to minimize the gap between scientific research and adoption of innovations into farm practice. I think this concept should be made available to a wider scientific audience as it may serve as a basis for further projects and approaches on different topics of animal health and welfare (probably, an adaption to the companion animal sector might also be possible).

However, there are some minor items that have to be addressed:

General:

Throughout the manuscript, you switch between writing numbers in figures (e.g. L. 218 “…3 networks included…”) or words (e.g. L. 216 “…networks varied from three to…”). However, I do not know if animals has a house style concerning this but please be consistent.

You should check the whole manuscript for spaces carefully (e.g. L. 268; L. 270)

Please add detailed information on author contribution.

Line remarks:

Introduction & Hennovation project:

L. 62: “…(p.390)…”

L. 66: Who is “many”? Citizens, consumers, scientists etc.? Please specify.

L. 72: Remove “,”.

L. 108: Which other actors?

Ll. 127-132: Very long and cumbersome phrase, suggest splitting it up in two sentences.

Materials & Methods:

L. 153: Did you mean “solutions”?

L. 154: “…facilitators’…”

L. 160: “…livestock sector; some had…”

L. 190: “…(Table 3)…”

L. 199: “…[11]. It…”

L. 201: “…Mathe et al. (p9)…”

Results:

L. 216: “…(Table 2)…”

L. 223: Who is meant by “those”? The actors?

L. 232: “…’soft’…”

L. 244: “…(Table 3)…”

L. 275: “…(Table 2)…”

L. 277: “…certification inspectors…”

L. 316: Please specify “…Welfare Quality parameters…”. Do you refer to the Welfare Quality® protocol(s) or to welfare indicators in general?

L. 317: “…presence of red mite in…”? Add “the stable/the hen house” or remove “in”.

L. 326: “…practice-led…”

Discussion:

L. 328-329: What food security challenges will we have to face in the future? Please specify. Are there also animal health/welfare challenges/changes in consumers’ perceptions?

L. 361-369: I think this paragraph would fit better in the results section.

L. 372-374: Please provide a reference for this statement.

L. 414: You should discuss the relatively low levels of dissemination (Table 3, Step 6) of the innovations from the Hennovation project. Which were the reasons for these findings? Were there not enough resources/efforts/time yet to implement the innovations into a broader practice? Please also discuss: What happened to the networks after the end of the projects? Are they still active? Do they have the potential to be active in 5-10 years? What would be needed to ensure that these networks will be active in the future?

Table 1: Can you provide a reference for the “Healthy Feet Project”?

Table 2: Please explain the abbreviations of the different networks. The first letters are clearly the abbreviation for the countries in which the networks were established but what do the letters after the minus stand for? At least explain that these are the specific acronyms/names of the projects.

I think it would be very interesting and helpful to add a column with references to already available publications that resulted from each network.

Table 3: This table adds valuable background information. However, it is very large. Maybe it would be possible to include it as supplementary material?

References:

Please check the references carefully, and provide online resources (links) wherever possible (also for research reports etc.).

Author Response

The authors appreciate the positive response from the reviewer, who clearly sees the manuscript as worthwhile. We have taken on board the reviewer's comments and the following corrections have been made:

Point 1. Specific comments made by the reviewer related to the switching between writing numbers in figures and words, spacing, references, all have been corrected (L62, L72, L127-132, L153, L154, L160, L190, L199, L201, L216, L232, L244, L275, L 277, L317, L326, L372 -374, reference Table 1. 

Further responses to specific comments:

Point 2, Reviewer’s Comment L. 66: Who is “many”? Citizens, consumers, scientists etc.?  Authors’ response: this section has been deleted to make the writing clearer and text more concise.

Point 3, Reviewer’s Comment L.223: Who is meant by “those”? The actors? Authors’ response: This sentence has been clarified and now reads as follows ‘The four end-of-lay networks included actors (such as production, catcher and abattoir managers, hen processors, handling equipment manufacturers, industry organisation representatives) actively involved in the process of depopulating, transporting and use of hens at the end of their laying period.’

Point 4, Reviewer’s Comment L. 316: Please specify “…Welfare Quality parameters…”. Do you refer to the Welfare Quality® protocol(s) or to welfare indicators in general? Authors’ response: Welfare Quality® protocol(s) parameters, this has been clarified in the text.

Point 5, Reviewer’s Comment L328-329: What food security challenges will we have to face in the future? Please specify. Are there also animal health/welfare challenges/changes in consumers’ perceptions? Authors’ response: This sentence has been rewritten to clarity the reviewers point and now reads as follows: Promoting innovation in agriculture is and will remain a major policy priority considering the emerging challenges of climate change, anti-microbial resistance and changing consumer demand in a growing population.’

Point 6, Reviewer’s Comment L.361-369: I think this paragraph would fit better in the results section. Authors’ response: Thank you for pointing this out, we have moved this to the result section.

Point 7, Reviewer’s Comment L.414: You should discuss the relatively low levels of dissemination (Table 3, Step 6) of the innovations from the Hennovation project. Which were the reasons for these findings? Were there not enough resources/efforts/time yet to implement the innovations into a broader practice? Please also discuss: What happened to the networks after the end of the projects? Are they still active? Do they have the potential to be active in 5-10 years? What would be needed to ensure that these networks will be active in the future? Authors’ response: There was a low level of dissemination as not all networks had finished the innovation process when this data was recorded by May 2017 (only 18% of all the networks had reached to step 6 at that stage). And as mentioned in the text ‘Despite the relatively short time scale of the project (30 months) seven networks actively sought to disseminate innovation beyond network members.’ At the end of the project nine networks wished to continue further trials with a further two networks keen to seek further financial support for trial work beyond the end date of the Hennovation project. Further details on facilitating these networks and requirements for these networks to run beyond the lifespan of the project e.g. professional facilitation, moderate resource support and access to relevant expertise, can be found in [28] & [36].

Point 8, Reviewer’s Comment Table 2: Please explain the abbreviations of the different networks. The first letters are clearly the abbreviation for the countries in which the networks were established but what do the letters after the minus stand for? At least explain that these are the specific acronyms/names of the projects. Authors’ response: The authors thank the reviewer for pointing this out. The letters refer to a specific network in a country. As this is not relevant information for the content of this manuscript, we have changed this to indicate the location in terms of country only.

Point 9, Reviewer’s Comment I think it would be very interesting and helpful to add a column with references to already available publications that resulted from each network. Authors’ response: As these innovation processes where not led by researchers but supported by researchers and implemented by farmers, no scientific publications have been written. Several articles have been published in farmer and poultry related professional magazines.

Point 10, Reviewer’s Table 3: This table adds valuable background information. However, it is very large. Maybe it would be possible to include it as supplementary material? Authors’ response: The large table has been added as supplementary material and a reduced version of this table has been included as Table 3. The information from the large table has also been presented as two new figures, Figure 1 & 2.

Point 11, Reviewer’s Comment: Please check the references carefully, and provide online resources (links) wherever possible (also for research reports etc.). Authors’ response: All references have been checked and online links added where available.

Reviewer 3 Report

The paper is interesting nevertheless there are some minor problem that authors should addressed.

GENERAL

The page numbering is incorrect

The tables must be numbered based on their presentation in the text

Table 3: please verify the allignment. In step 3 the sum of the first line is 15 instead of 19 why?

For other comments please see the attached text

Author Response

The authors appreciate the positive response from the reviewer, who clearly sees the manuscript as interesting. We have taken on board the reviewer's comments and the following corrections have been made:

Point 1. Specific comments made by the reviewer, related to the page numbering, table number in the text and in-text references, all have been corrected.

Further responses to specific comments:

Point 2, Reviewer’s Comment Table 3: please verify the alignment. In step 3 the sum of the first line is 15 instead of 19 why? Authors’ response: The authors thank the reviewer for noticing this. The alignment has been adjusted and the error corrected.

Point 3, Reviewer’s Comment L256: From this table it is not clear that only one network completed the step 3. Please explain better. Authors’ response: All networks apart from one network completed step 3. To make sure this sentence is clear. it has been rewritten and now reads as following: Analysis of the progress of each network on the six steps shows that in project month 29 (May 2017), all networks were active, with over 90% (n=19) completing the action planning (step 3, see Figure 2).

Point 4, Reviewer’s Comment L258: Where is this % reported? Authors’ response: The authors reviewed this sentence and this information is available in the table (now supplementary material). We have deleted this sentence to prevent duplication and confusion.

Point 5, Reviewer’s Comment L304: In my opinion this paragraph L304-L326 should be shifted in 4.2 Authors’ response: This paragraph has been moved to section 4.2 as suggested by the reviewer.

Reviewer 4 Report

I enjoyed reading the manuscript about the project HENNOVATION and I agree with the broad ideas of the authors that interactions with all stakeholders – farmers, industry, scientists help defining study areas and can lead to innovative ideas. I would add that the willingness to implement a new idea is greater if the person perceives this idea as his/her own. This could be an additional advantage of the approach because all the stakeholders can identify themselves better with the innovations.

There are not many systematic reports on how networks of stakeholders can improve animal welfare in laying hens so this publication is important.

However, very little is reported on the impact on the laying hens. It would be helpful to apply the concept input – output – impact. This manuscript describes the input and the output of the hennovation project. A possible impact could be that farmers report less problems with feather-pecking, that the mortality rate is going down, productivity rises or something of this sort. This is not mentioned. Are there plans to collect data on that or has this been done already? This would be important to judge the concept of hennovation.

Overall, the text is easy to follow. The few following suggestions are meant to clarify certain points.

Can you define what is called an innovation? Does it have to included animal welfare, productivity, or both?

The procedure beyond stating a problem and possible solutions on part of the scientists is not detailed. Are hypotheses generated? Are studies conducted to test these hypotheses? Are these on small experimental units, are they done on commercial farms, in which way?

Line 114: In the EU strategy ‘Europe 2020’ citation animal welfare is not mentioned. Was it intended to include animal welfare or is animal welfare part of this strategy?

Lines 148ff: It would be helpful to have a figure showing the hierarchical structure of coordinators and facilitators.

Table 3: How were the data in Table 3 generated? Was there a survey or structured interview? With closed questions? Were these terms from the table in those questions?

Line 252: Can you name examples of facilitator interventions?

Line 330: This claim is very important for the project but it remains rather imprecise. Alpacas are not a good example for hennovation. Can you name more specific examples related to laying hens?

Author Response

The authors appreciate the positive response from the reviewer, who clearly sees the manuscript as important. We have taken on board the reviewer's comments and the following corrections have been made:

Point 1, Reviewer’s Comment However, very little is reported on the impact on the laying hens. It would be helpful to apply the concept input – output – impact. This manuscript describes the input and the output of the Hennovation project. A possible impact could be that farmers report less problems with feather-pecking, that the mortality rate is going down, productivity rises or something of this sort. This is not mentioned. Are there plans to collect data on that or has this been done already? This would be important to judge the concept of Hennovation. Authors’ response: Additional reference has been made to impact e.g. ‘Where trial work was undertaken, relevant outcome measures were used to assess the impact of innovations. These included productivity parameters (e.g. egg production and mortality), welfare measures (e.g. feather scores and Welfare Quality® protocol(s) parameters) and environment indicators (e.g. ammonia and presence of red mite).’ A full description of impact assessment is beyond scope of this paper.

Point 2, Reviewer’s Comment: Can you define what is called an innovation? Does it have to included animal welfare, productivity, or both? Authors’ response: The Hennovation project focused on improvement of the health and welfare of laying, specifically on finding practical solutions to problems related to feather (or injurious) pecking on-farm and on the handling and use of end-of-lay hens off-farm. A broad definition of innovation was used to include application of both novel and existing ideas or best practices in new circumstances to improve laying hen health and welfare (see first paragraph of section 2 The Hennovation project).  

Point 3, Reviewer’s Comment: The procedure beyond stating a problem and possible solutions on part of the scientists is not detailed. Are hypotheses generated? Are studies conducted to test these hypotheses? Are these on small experimental units, are they done on commercial farms, in which way? Authors’ response:  Thank you for asking further clarification about this point. We have added additional information to make this clearer L178: ‘For example the practical trialling and development description (step 4) could have included small scale experimental trials or a more informal feasibility trial.’

Point 4, Reviewer’s Comment Line 114: In the EU strategy ‘Europe 2020’ citation animal welfare is not mentioned. Was it intended to include animal welfare or is animal welfare part of this strategy? Authors’ response: Animal welfare is included, and part of this strategy, as improved animal welfare invariably is accompanied with sustainability and competitiveness.

Point 5, Reviewer’s Comment Lines 148ff: It would be helpful to have a figure showing the hierarchical structure of coordinators and facilitators. Authors’ response: The authors reviewed this comment, however, perceive this is beyond the scope of this manuscript. The full project proposal is available from https://cordis.europa.eu/project/rcn/194800_en.html.

Point 6, Reviewer’s Comment Table 3: How were the data in Table 3 generated? Was there a survey or structured interview? With closed questions? Were these terms from the table in those questions? Authors’ response: The framework in the table (previous table 3) was developed by and with facilitators during the first facilitators’ workshop as a tool to guide the facilitation of the innovation process, record level of engagement (Figure 1 and 2) and to stimulate facilitator learning in managing the process in the field. In addition to individual reflection by each network facilitator, participants were also asked to compare the progress and functioning of each network and identify why similarities or differences in performance emerged. Further clarification of this has been provided in section 3 materials and methods L119 -L183 and in L232 further insight is provided at the specific data collection points as the facilitators used the framework to reflect on the functioning of their networks as they moved through the innovation process.

Point 7, Reviewer’s Comment Line 252: Can you name examples of facilitator interventions? Authors’ response: Examples have been added L239-241‘Examples of interventions required included the involvement of a scientist with specific knowledge on a problem identified by a network to jointly find solutions to in step 1 or the involvement of farm business advisor to assess the feasibility of an idea developed in step 3.’

Point 8, Reviewer’s Comment Line 330: This claim is very important for the project, but it remains rather imprecise. Alpacas are not a good example for Hennovation. Can you name more specific examples related to laying hens? Authors’ response: The authors reviewed this comment and do perceive, however, the use of Alpacas by the organic family farms in Spain as good example of an innovation generated by an on-farm innovation network in a specific context. The producers involved in this network reported problems of high mortality levels (up to 30% mortality rate) that are unsustainable due to predation, killing of free-range hens by birds of prey. The producers reported that birds of prey, such as the common buzzard, were particularly difficult to control as they hunt during the day when the hens are outside, and their attacks are unpredictable. Moreover, the common buzzard is a widespread and a legally protected species. The use of alpacas led to a reduced mortality and stress, and besides being a very significant economic problem, predation can originate social conflicts between authorities and producers claiming for their animals’ loss.